# Customized Multiple Clustering via Multi-Modal Subspace Proxy Learning

**Jiawei Yao**[1]     **Qi Qian**[2*]     **Juhua Hu**[1†]

[1] School of Engineering and Technology, University of Washington, Tacoma, WA 98402, USA
[2] Zoom Video Communications
{jwyao, juhuah}@uw.edu, qi.qian@zoom.us

## Abstract

Multiple clustering aims to discover various latent structures of data from different aspects. Deep multiple clustering methods have achieved remarkable performance by exploiting complex patterns and relationships in data. However, existing works struggle to flexibly adapt to diverse user-specific needs in data grouping, which may require manual understanding of each clustering. To address these limitations, we introduce Multi-Sub, a novel end-to-end multiple clustering approach that incorporates a multi-modal subspace proxy learning framework in this work. Utilizing the synergistic capabilities of CLIP and GPT-4, Multi-Sub aligns textual prompts expressing user preferences with their corresponding visual representations. This is achieved by automatically generating proxy words from large language models that act as subspace bases, thus allowing for the customized representation of data in terms specific to the user's interests. Our method consistently outperforms existing baselines across a broad set of datasets in visual multiple clustering tasks. Our code is available at https://github.com/Alexander-Yao/Multi-Sub.

## 1   Introduction

Clustering is a fundamental technique for analyzing data based on certain similarities, attracting extensive attention due to the abundance of unlabeled data. Traditional clustering methods [MacQueen et al., 1967, Ng et al., 2001, Bishop and Nasrabadi, 2006] rely on general-purpose handcrafted features that may not suit specific tasks well. Deep clustering algorithms have improved clustering performance by employing Deep Neural Networks (DNNs) [Xie et al., 2016, Guérin and Boots, 2018, Qian et al., 2022, Qian, 2023] to learn task-specific features. However, most of these algorithms assume a single partition of the data, while real data can be clustered differently according to different aspects, e.g., fruits in Fig. 1 can be grouped differently by color or by species.

Multiple clustering algorithms [Bae and Bailey, 2006, Hu et al., 2017] address this challenge by producing multiple partitions of the data for various applications,

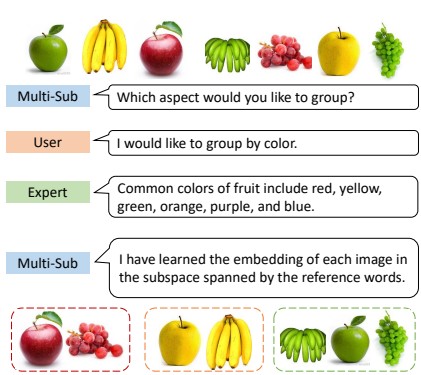

Figure 1: The workflow of Multi-Sub that obtains a desired clustering based on the subspace spanned by reference words obtained from GPT-4 using users' high-level interest.

---

*Work done while at Alibaba Group.
†Corresponding author

showing the capability of discovering multiple clusterings from a single dataset. For instance, in e-commerce, products can be clustered by category for inventory management or by customer preferences for personalized recommendations. Recently, there has been a growing interest in incorporating deep learning techniques into multiple clustering. These techniques mainly use auto-encoders and data augmentation methods to extract a wide range of feature representations, which enhance the quality of multiple clustering [Miklautz et al., 2020, Ren et al., 2022, Yao et al., 2023].

For real-world applications, a key challenge for end users is efficiently identifying the desired clustering from multiple results based on their interests or application purposes. We observe that users are willing to indicate their interest using succinct keywords (e.g., color or species for fruits in Fig. 1). However, it is difficult to use only a concise keyword to directly extract the corresponding image representations. Fortunately, the recent development of multi-modal models like CLIP [Radford et al., 2021] that align images with their text descriptions can help bridge this gap. Nevertheless, unlike methods that can use labeled data to fine-tune pre-trained models [Gao et al., 2023, Wang et al., 2023] to learn new task-specific representations, multiple clustering often faces scenarios with ambiguous or unspecified label categories and quantities. Therefore, given only a high-level concept from the user, it is intractable to fine-tune pre-trained models to capture a particular aspect of the data in an unsupervised manner. Very recently, Multi-MaP [Yao et al., 2024] leverages CLIP to learn textual and image embeddings simultaneously that follow the user's high-level textual concept. However, to achieve better performance, they require the user to provide a contrastive concept that is different from the desired concept, which may not be feasible in many real-world applications. Moreover, they obtain the new representations at first and then apply the traditional clustering method like k-means in a separate stage. This insufficient optimization lacking refinement between stages makes the clustering performance sub-optimal.

To mitigate these challenges, in this work, we first assume that the desired image and textual representations are residing in the same subspace according to the user's specific concept. Thereafter, to capture the desired subspace better, we can ask low-cost experts like Google or large language models (LLMs) (e.g., GPT-4) for common categories under the desired concept, as illustrated in Fig. 1. Although those returned common categories may not directly capture the clustering targets, they can be applied as the subspace basis to help search the appropriate representations inside. More importantly, during the learning under the desired subspace, we also incorporate the clustering loss to learn the representations and obtain the clustering simultaneously, which significantly enhances the model's clustering performance and efficiency. The main contributions of this work can be summarized as follows.

- We present a novel multiple clustering method, Multi-Sub, that can explicitly capture a user's clustering interest by aligning the textual interest with the visual features of images. Concretely, we propose to learn the desired clustering proxy in the subspace spanned by the common categories under a user's interest.

- Unlike most existing multiple clustering methods that require distinct stages for representation learning and clustering, Multi-Sub can obtain both the desired representations and clustering simultaneously, which can significantly improve the clustering performance and efficiency.

- Extensive experiments on all publicly available multiple clustering tasks empirically demonstrate the superiority of the proposed Multi-Sub, with a precise capturing of a user's interest.

## 2 Related Work

### 2.1 Multiple Clustering

Multiple clustering, a methodology capable of unveiling alternative data perspectives, has garnered significant interest. Traditional approaches for multiple clustering [Hu and Pei, 2018] employ shallow models to identify diverse data groupings. Some methods, such as COALA [Bae and Bailey, 2006] and [Qi and Davidson, 2009], utilize constraints to generate alternative clusterings. Other techniques leverage distinct feature subspaces to produce multiple clusterings, as exemplified by [Hu et al., 2017] and MNMF [Yang and Zhang, 2017]. Information theory has also been applied to generate multiple clusterings, as demonstrated by [Gondek and Hofmann, 2003] and [Dang and Bailey, 2010].

Recent advancements have seen the application of deep learning to discover multiple clusterings, yielding improved clustering performance. For instance, [Wei et al., 2020] proposed a deep matrix factorization method that utilizes multi-view data to identify multiple clusterings. ENRC [Miklautz et al., 2020] employs an auto-encoder to learn object features and optimizes a clustering objective function to find multiple clusterings. iMClusts [Ren et al., 2022] leverages auto-encoders and multi-head attention to learn features from various perspectives and discover multiple clusterings. AugDMC [Yao et al., 2023] uses data augmentation to generate diverse image aspects and learns representations to uncover multiple clusterings. DDMC [Yao and Hu, 2024] employs a variational Expectation-Maximization framework with disentangled representations to achieve superior clustering outcomes. However, almost all of these methods necessitate substantial user efforts to understand and select the appropriate clustering for different application purposes. Recently, Multi-MaP [Yao et al., 2024] leverages CLIP encoders to align a user's interest with visual data by learning representations close to the interested concept but far away from a contrastive concept, significantly improving the efficiency of capturing user-desired clusterings. However, Multi-MaP requires the user to input a contrastive concept for better performance, which is often not applicable. More importantly, it separates the representation learning and clustering as two distinct stages, which may result in sub-optimal performance. These issues will be mitigated in this work.

## 2.2 Multi-Modal Models

Multi-modal learning involves acquiring representations from various input modalities like image, text, or speech. Here, we focus on how vision models benefit from natural language supervision. A key model in this area is CLIP [Radford et al., 2021], which aligns images with their corresponding text using contrastive learning on a dataset of 400 million text-image pairs.

Fine-tuning adapts vision-language models, such as CLIP, for specific image recognition tasks. This is seen in CoOp [Zhou et al., 2022] and CLIP-Adapter [Gao et al., 2023], the latter using residual style feature blending to enhance performance. TeS [Wang et al., 2023] highlights the efficacy of fine-tuning in improving visual comprehension through natural language supervision. With limited labeled data, zero-shot learning has gained attention. Some approaches surpass CLIP by integrating other large pre-trained models. For example, VisDesc [Menon and Vondrick, 2022] uses GPT-3 to generate contextual descriptions for class names, outperforming basic CLIP prompts. UPL [Huang et al., 2022] and TPT [Shu et al., 2022] utilize unlabeled data to optimize text prompts. InMaP [Qian et al., 2024] and the online variant [Qian and Hu, 2024] aid class proxies in vision space with text proxies. Recent advancements have significantly improved vision-language pre-training using large-scale noisy datasets. ALIGN [Jia et al., 2021] employs over one billion image alt-text pairs without expensive filtering, showing that corpus scale can offset noise. Similarly, BLIP-2 [Li et al., 2023] uses a novel framework to bootstrap captions from noisy web data, enhancing both vision-language understanding and generation tasks. While these methods strive to enhance the performance of vision classification tasks, clustering presents a distinct scenario where class names are not available to extract useful information from multi-modal information as in this work.

## 3 The Proposed Method

Given a dataset of images $\{x_i\}_{i=1}^{n}$ and user-defined preferences for data grouping (such as color and species), our goal is to generate clustering results that are specifically tailored to each preference. Thereafter, end users can directly use them for different application purposes without additional manual selection efforts. This process poses significant challenges, as it requires accurately aligning the complex, multi-dimensional data of images with the subjective and varied textual preferences of users. Traditional clustering methods often fail to capture these nuances, leading to a generic and less informative categorization for specific user applications.

Recently, the CLIP model [Radford et al., 2021] facilitated a more natural alignment between textual interests and visual representations. Our method, Multi-Sub, extends this alignment through a novel multi-modal subspace proxy learning approach. Fig. 2 outlines the overall framework of Multi-Sub, which is tailored to capture and respond to the diverse interests of users in clustering tasks. Multi-Sub employs a two-phase iterative approach to align and cluster images based on user-defined preferences such as color and species as described below.

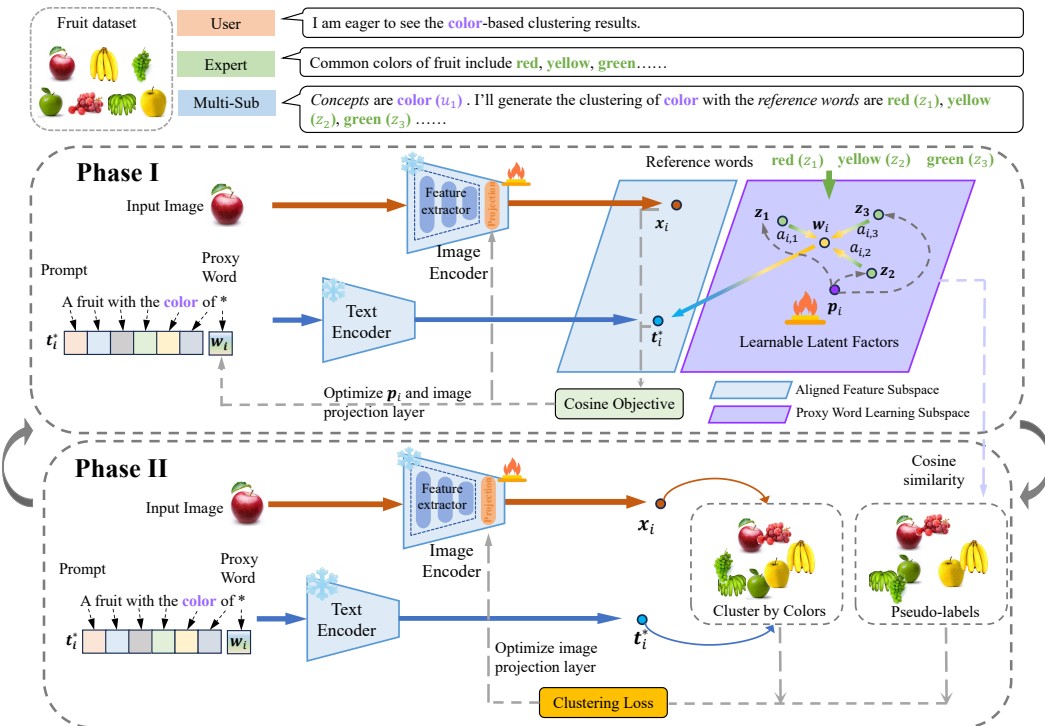

Figure 2: **Multi-Sub framework.** In Multi-Sub framework, **Phase I** (Proxy Learning and Alignment) processes each image $x_i$ with user-defined textual prompts through a partially learnable image encoder (with a learnable projection layer) and a frozen text encoder. The latent factor $\mathbf{p}_i$ calculates weights $\{a_{i,k}\}_{k=1}^K$ based on the similarity to reference word embeddings $\{\mathbf{z}_i\}_{k=1}^K$, which are then aggregated to form the proxy word embedding $\mathbf{w}_i$. This proxy word embedding, combined with the image representation $\mathbf{x}_i$, establishes the Aligned Feature Subspace for better alignment between the text and image under the user's interest. In **Phase II** (Clustering), given the learned proxy word embeddings $\{\mathbf{w}_i\}$ from Phase I to form pseudo-labels, the projection layer of the image encoder is further refined using the clustering loss. In Phase I, both the latent factor $\mathbf{p}$ and the projection layer learn 100 epochs, after which the projection layer further learns 10 epochs using the clustering loss in Phase II. This alternative process repeats until convergence.

## 3.1 Background: Multi-Modal Pre-Training in CLIP

Let $\{x_i, t_i\}_{i=1}^n$ be a set of image-text pairs, where $x_i$ denotes an image and $t_i$ denotes its corresponding text description. We can obtain the vision and text representations of each pair by applying two encoders, $f(\cdot)$ and $h(\cdot)$, as $\mathbf{x}_i = f(x_i)$ and $\mathbf{t}_i = h(t_i)$. Both $f(\cdot)$ and $h(\cdot)$ are encoders that optimize the vision and text representations, respectively, such that $\mathbf{x}_i$ and $\mathbf{t}_i$ are unit vectors. The primary goal during this pre-training phase is to minimize the contrastive loss, formulated as

$$\min_{f,h} \sum_i - \log \frac{\exp(\mathbf{x}_i^\top \mathbf{t}_i / \tau)}{\sum_j \exp(\mathbf{x}_i^\top \mathbf{t}_j / \tau)} - \log \frac{\exp(\mathbf{t}_i^\top \mathbf{x}_i / \tau)}{\sum_j \exp(\mathbf{t}_i^\top \mathbf{x}_j / \tau)} \tag{1}$$

where $\tau$ is a temperature parameter. The contrastive loss encourages the alignment of the image and its description while penalizing the similarity of the image with irrelevant texts [Qian et al., 2019]. The efficacy of this contrastive approach is vital for the subsequent phases of proxy word learning and fine-grained clustering, as it ensures that the foundational embeddings accurately reflect the inherent content and context of each modality.

## 3.2 Subspace Proxy Word Representation

We build upon the pre-trained image and text encoders from CLIP and investigate whether we can leverage the image-text alignment to extract user-specific information. Specifically, given a fruit

image [Hu et al., 2017] as illustrated in Fig. 2, different users may have different interests of its attributes, such as color, species, etc. However, the pre-trained image encoder in CLIP can only produce a single image embedding, which may not capture a user's interest exactly, not mentioning capturing different aspects. Furthermore, unlike classification tasks, clustering tasks do not come with concrete cluster names or numbers. Therefore, we cannot directly use the pre-trained text encoder of CLIP to generate the corresponding text embedding.

To address these challenges, we propose a subspace proxy word learning method to learn new embedding under the preferred aspect provided by the user. Thereafter, the main challenge is, given only a high-level concept like 'color' as in Fig. 2, how to effectively represent its subspace. Since the high-level concept itself cannot reflect different details under this concept in different images, it is difficult to do effective alignment between the high-level concept and images to figure out the corresponding vision subspace. Therefore, we propose to figure out the text subspace at first. Concretely, given pre-trained large language models like GPT-4 as low-cost experts, we can quickly gather common categories under a high-level concept using only one query like 'what are the common fruit colors' in Fig. 2. However, we cannot directly use the returned categories to do grouping, since they may not cover all existing categories in the data. Instead, we consider that most categories in the data under this concept are residing in the same subspace as the returned ones. Therefore, we can apply suggested categories as basis or reference words in the subspace. Then, each image's category under the desired concept can be represented by a linear combination of these reference words.

Assuming GPT-4 provides $K$ reference words as $\{z_k\}_{k=1}^K$, the proxy word of image $x_i$ can be calculated as

$$\mathbf{w}_i = \sum_{k=1}^K a_{i,k}\phi(z_k) \tag{2}$$

where $\phi(z_k)$ is the token embedding of reference word $z_k$ and $\{a_{i,k}\}_{k=1}^K$ are weights corresponding to each reference word as a basis. A higher weight $a_{i,k}$ indicates that the image $x_i$'s category is closer to the reference word $z_k$. Here, we introduce trainable latent factor $\mathbf{p}_i$ to learn the weight $a_{i,k}$, and it can be calculated as

$$a_{i,k} = \frac{\exp\left(\mathbf{p}_i\mathbf{z}_k\right)}{\sum_j \exp\left(\mathbf{p}_i\mathbf{z}_j\right)} \tag{3}$$

where $\mathbf{z}_k = \phi(z_k)$. Thereafter, $\mathbf{w}_i$ is representing the token embedding of image $x_i$'s proxy word under the preferred user concept. Once $\mathbf{p}_i$ is well obtained, the image's proxy word representation under the preferred user concept is also obtained. Next, we discuss how to learn $\mathbf{p}_i$ using CLIP.

### 3.3 Multi-Modal Subspace Proxy Learning

As mentioned above, CLIP's text and image encoders were learned by aligning the text prompt with its corresponding image. The standard text prompt of CLIP is designed as "a photo of a fruit" for an image containing "fruit". Now, given a user's preference (e.g., color), we can rewrite the prompt as "a fruit with the color of *" denoted by $t_i^*$ for image $x_i$, where "*" is the placeholder for the unknown proxy word of image $x_i$ under concept 'color' and its token embedding $\mathbf{w}_i$ can be formulated as the linear superposition of reference words' token embeddings as discussed above.

Thereafter, the prompt text embedding after the text encoder can be formulated as

$$\mathbf{t}_i^* = h(\phi(t_i^*)\|\phi(w_i)) \tag{4}$$

To effectively learn $\mathbf{p}_i$, the trainable latent factors, we utilize the alignment capabilities of CLIP by adjusting these factors so that the weighted sum of reference word embeddings closely aligns with the visual representation of the image. This process involves iteratively adjusting $\mathbf{p}_i$ to maximize the cosine similarity between the image's representation $\mathbf{x}_i$ and its corresponding proxy word embedding $\mathbf{w}_i$. The optimization is conducted with the following loss function:

$$\mathcal{L}(\mathbf{w}_i) = -\langle f(x_i), h(\phi(t_i^*)\|\phi(w_i))\rangle \tag{5}$$

It should be noted that this optimization procedure can be conducted with both the text encoder and image encoder frozen, which is very efficient. However, the image embedding extracted directly from the pre-trained image encoder may not reflect its representation under the desired user interest. Therefore, during the optimization procedure, we do freeze the text encoder but open the image encoder. Nevertheless, to preserve the strong capacity of the pre-trained image encoder in CLIP, we open only the projection layer of the image encoder, while its remaining parameters are frozen as shown in the 'Phase I' of Fig. 2.

## 3.4 Clustering Loss

To enhance the clustering performance of Multi-Sub, in 'Phase II', we leverage pseudo-labels assigned using the currently learned proxy word embeddings $\{\mathbf{w}_i\}$ and image embeddings $\{\mathbf{x}_i\}$ from 'Phase I'. Concretely, each image $x_i$ can be represented by the concatenation of its currently learned proxy word embedding $\mathbf{w}_i$ and image embedding $\mathbf{x}_i$, denoted as $\mathbf{v}_i = [\mathbf{w}_i, \mathbf{x}_i]$. The pseudo-labels can be obtained by an offline k-means on $\{\mathbf{v}_i\}$, which is however not efficient. Considering that proxy words for data points within the same cluster should show similar relationships to reference words, we obtain the pseudo-labels using the highest cosine similarity between the currently learned proxy word embeddings $\{\mathbf{w}_i\}$ and the reference word embeddings $\{\mathbf{z}_k\}$.

Given the pseudo-labels, the image embeddings can be further optimized by opening only the projection layer of the image encoder for improved compactness and separability in clusters. This loss consists of two primary components: intra-cluster loss and inter-cluster loss, aimed at refining cluster cohesion and separation, respectively. It should be noted that to better represent each image under the desired user concept, we define the clustering loss over $\mathbf{v}_i$ containing both textual and visual information.

**Intra-cluster Loss:** The intra-cluster loss is designed to minimize the distances between embeddings within the same cluster, encouraging cluster compactness. It is calculated using the following formula:

$$\mathcal{L}_{\text{intra}} = \frac{1}{N_{\text{intra}}} \sum_{i,j \in \text{intra}} \|\mathbf{v}_i - \mathbf{v}_j\|^2 \tag{6}$$

Here, $\|\mathbf{v}_i - \mathbf{v}_j\|^2$ is the squared Euclidean distance between embeddings $\mathbf{x}_i$ and $\mathbf{x}_j$ of data points $i$ and $j$ within the same cluster, and $N_{\text{intra}}$ denotes the number of intra-cluster pairs.

**Inter-cluster Loss:** This component aims to maximize the distances between embeddings from different clusters, thus enhancing separability. The inter-cluster loss is defined by a margin-based hinge loss as follows:

$$\mathcal{L}_{\text{inter}} = \frac{1}{N_{\text{inter}}} \sum_{i,j \in \text{inter}} \max(0, m - \|\mathbf{v}_i - \mathbf{v}_j\|) \tag{7}$$

where $\max(0, m - \|\mathbf{v}_i - \mathbf{v}_j\|)$ computes the hinge loss for each pair of embeddings from different clusters, ensuring a minimum margin $m$ between them. $N_{\text{inter}}$ is the count of inter-cluster pairs.

**Total Loss:** The overall clustering loss combines the intra- and inter-cluster losses, moderated by a balancing factor $\lambda$:

$$\mathcal{L}_{\text{total}} = \lambda \cdot \mathcal{L}_{\text{intra}} + (1 - \lambda) \cdot \mathcal{L}_{\text{inter}} \tag{8}$$

Optimizing this loss function in 'Phase II' helps regularize the embedding space where clusters are both internally dense and well-separated from each other. It should be noted that in this phase we aim to learn a better projection layer only for the image encoder, while all others are fixed as shown in 'Phase II' of Fig. 2.

Previous methods often use a two-stage strategy that separates representation learning and clustering to simplify the optimization process. This separation, however, can lead to sub-optimal clustering results, since the learned representations may not be fully aligned with the clustering objective without refinement. In this work, we obtain both the proxy word and the clustering alternatively and simultaneously. Concretely, we first learn the proxy word in a user-preferred subspace. Then, we fix the proxy word and refine the image encoder further to obtain better image representations using the clustering objective. These two phases are repeated alternatively until convergence, where 'Phase I' learns 100 epochs and 'Phase II' learns 10 epochs in each alternating according to the empirical experience as summarized in Fig. 2.

Table 1: Dataset Statistics.

| Datasets | # Samples | # Hand-crafted features | # Clusters |
|---|---|---|---|
| Standford Cars | 1,200 | wheelbase length; body shape; color histogram | 4;3 |
| Card | 8,029 | symbol shapes; color distribution | 13;4 |
| CMUface | 640 | HOG; edge maps | 4;20;2;4 |
| Fruit | 105 | shape descriptors; color histogram | 3;3 |
| Fruit360 | 4,856 | shape descriptors; color histogram | 4;4 |
| Flowers | 1,600 | petal shape; color histogram | 4;4 |
| CIFAR-10 | 60,000 | edge detection; color histograms; shape descriptors | 2;3 |

Table 2: Quantitative comparison. The significantly best results with 95% confidence are in bold.

| Dataset | Clustering | MSC NMI↑ | MSC RI↑ | MCV NMI↑ | MCV RI↑ | ENRC NMI↑ | ENRC RI↑ | iMClusts NMI↑ | iMClusts RI↑ | AugDMC NMI↑ | AugDMC RI↑ | DDMC NMI↑ | DDMC RI↑ | Multi-MaP NMI↑ | Multi-MaP RI↑ | Multi-Sub NMI↑ | Multi-Sub RI↑ |
|---|---|---|---|---|---|---|---|---|---|---|---|---|---|---|---|---|---|
| Fruit | Color | 0.6886 | 0.8051 | 0.6266 | 0.7685 | 0.7103 | 0.8511 | 0.7351 | 0.8632 | 0.8517 | 0.9108 | 0.8973 | 0.9383 | 0.8619 | 0.9526 | **0.9693** | **0.9964** |
| | Species | 0.1627 | 0.6045 | 0.2733 | 0.6597 | 0.3187 | 0.6536 | 0.3029 | 0.6743 | 0.3546 | 0.7399 | 0.3764 | 0.7621 | 1.0000 | 1.0000 | **1.0000** | **1.0000** |
| Fruit360 | Color | 0.2544 | 0.6054 | 0.3776 | 0.6791 | 0.4264 | 0.6868 | 0.4097 | 0.6841 | 0.4594 | 0.7392 | 0.4981 | 0.7472 | 0.6239 | 0.8243 | **0.6654** | **0.8821** |
| | Species | 0.2184 | 0.5805 | 0.2985 | 0.6176 | 0.4142 | 0.6984 | 0.3861 | 0.6732 | 0.5139 | 0.7430 | 0.5292 | 0.7703 | 0.5284 | 0.7582 | **0.6123** | **0.8504** |
| Card | Order | 0.0807 | 0.7805 | 0.0792 | 0.7128 | 0.1225 | 0.7313 | 0.1144 | 0.7658 | 0.1440 | 0.8267 | 0.1563 | 0.8326 | 0.3653 | 0.8587 | **0.3921** | **0.8842** |
| | Suits | 0.0497 | 0.3587 | 0.0430 | 0.3638 | 0.0676 | 0.3801 | 0.0716 | 0.3715 | 0.0873 | 0.4228 | 0.0933 | 0.6469 | 0.2734 | 0.7039 | **0.3104** | **0.7941** |
| CMUface | Emotion | 0.1284 | 0.6736 | 0.1433 | 0.5268 | 0.1592 | 0.6630 | 0.0422 | 0.5932 | 0.0161 | 0.5367 | 0.1726 | 0.7593 | 0.1786 | 0.7105 | **0.2053** | **0.8527** |
| | Glass | 0.1420 | 0.5745 | 0.1201 | 0.4905 | 0.1493 | 0.6209 | 0.1929 | 0.5627 | 0.1039 | 0.5361 | 0.2261 | 0.7663 | 0.3402 | 0.7068 | **0.4870** | **0.8324** |
| | Identity | 0.3892 | 0.7326 | 0.4637 | 0.6247 | 0.5607 | 0.7635 | 0.5109 | 0.8260 | 0.5875 | 0.8334 | 0.6360 | 0.8907 | 0.6625 | 0.9496 | **0.7441** | **0.9834** |
| | Pose | 0.3687 | 0.6322 | 0.3254 | 0.6028 | 0.2290 | 0.5029 | 0.4437 | 0.6114 | 0.1320 | 0.5517 | 0.4526 | 0.7904 | 0.4693 | 0.6624 | **0.5923** | **0.8736** |
| Stanford Cars | Color | 0.2331 | 0.6158 | 0.2103 | 0.5802 | 0.2465 | 0.6779 | 0.2336 | 0.6552 | 0.2736 | 0.7525 | 0.6899 | 0.8765 | 0.7360 | 0.9193 | **0.7533** | **0.9387** |
| | Type | 0.1325 | 0.5336 | 0.1650 | 0.5634 | 0.2063 | 0.6217 | 0.1963 | 0.5643 | 0.2364 | 0.7356 | 0.6045 | 0.7957 | 0.6355 | 0.8399 | **0.6616** | **0.8792** |
| Flowers | Color | 0.2561 | 0.5965 | 0.2938 | 0.5860 | 0.3329 | 0.6214 | 0.3169 | 0.6127 | 0.3556 | 0.6931 | 0.6327 | 0.7887 | 0.6426 | 0.7984 | **0.6940** | **0.8843** |
| | Species | 0.1326 | 0.5273 | 0.1561 | 0.6065 | 0.1894 | 0.6195 | 0.1887 | 0.6077 | 0.1996 | 0.6227 | 0.6148 | 0.8321 | 0.6013 | 0.8103 | **0.6724** | **0.8719** |
| CIFAR-10 | Type | 0.1547 | 0.3296 | 0.1618 | 0.3305 | 0.1826 | 0.3469 | 0.2040 | 0.3695 | 0.2855 | 0.4516 | 0.3991 | 0.5827 | 0.4969 | 0.7104 | **0.5271** | **0.7394** |
| | Environment | 0.1136 | 0.3082 | 0.1379 | 0.3344 | 0.1892 | 0.3599 | 0.1920 | 0.3664 | 0.2927 | 0.4689 | 0.3782 | 0.5547 | 0.4598 | 0.6737 | **0.4828** | **0.7096** |

## 4 Experiments

**Datasets**  To demonstrate the effectiveness of Multi-Sub, we evaluate the proposed method on almost all publicly available visual datasets commonly used in multiple clustering tasks Yu et al. [2024], including Stanford Cars Yao et al. [2024], Card Yao et al. [2023], CMUface Günnemann et al. [2014], Flowers Yao et al. [2024], Fruit Hu et al. [2017] and Fruit360 Yao et al. [2023]. **Stanford Cars** contains two different clustering types, one for car color (e.g., red, blue, black) and one for car type (e.g., sedan, SUV, convertible), comprising 1,200 annotated car images. **Card** includes 8,029 images of playing cards, with two clustering types: one based on rank (e.g., Ace, King, Queen) and another on suit (e.g., clubs, diamonds, hearts, spades). **CMUface** provides 640 facial images with clustering options for pose (e.g., front-facing, side-facing), identity, glasses (with/without), and emotion (e.g., happy, neutral, sad). **Flowers** comprises 1,600 flower images with two clustering types: one for color (e.g., red, blue, yellow) and another for species (e.g., iris, aster). **Fruit** includes 105 images of fruits with two clustering criteria: species (e.g., apples, bananas, grapes) and color (e.g., green, red, yellow). **Fruit360**, similar to the Fruit dataset, contains 4,856 images annotated for species (e.g., apple, banana, cherry) and color.

Additionally, we created a multiple clustering dataset from **CIFAR-10** Krizhevsky et al. [2009] by organizing the images into clusters based on type and environment. For type, the clusters are transportation and animals. For environment, the clusters are land, air, and water. The dataset characteristics about data size, handcrafted features, and cluster information are also summarized in Table 1.

It should be noted that some data may face challenges in extraction of meaningful candidate categories from GPT-4, or their labels lack semantic features. Taking the identity clustering on the CMUface dataset Günnemann et al. [2014] as an example, different identities correspond to different individuals, and the names' semantic meanings should not affect clustering outcomes. In such cases, following the Multi-Map setting Yao et al. [2024], we randomly select 10 words from WordNet Fellbaum [2010] as reference categories.

**Baselines**  We compare our Multi-Sub with seven state-of-the-art multiple clustering methods. These methods are: **MSC** Hu et al. [2017] is a traditional multiple clustering method that uses hand-crafted features to automatically find different feature subspace for different clusterings; **MCV** Guérin and Boots [2018] leverages multiple pre-trained feature extractors as different views of the same data; **ENRC** Miklautz et al. [2020] integrates auto-encoder and clustering objective to generate different clusterings; **iMClusts** Ren et al. [2022] is a deep multiple clustering method that leverages the expressive representational power of deep autoencoders and multi-head attention to generate

Table 3: Variants of CLIP. The significantly best results with 95% confidence are in bold.

| Dataset | Clustering | CLIP$_{GPT}$ | | CLIP$_{label}$ | | Multi-Sub | |
|---|---|---|---|---|---|---|---|
| | | NMI↑ | RI↑ | NMI↑ | RI↑ | NMI↑ | RI↑ |
| Fruit | Color | 0.7912 | 0.9075 | 0.8629 | 0.9780 | **0.9693** | **0.9964** |
| | Species | 0.9793 | 0.9919 | 1.0000 | 1.0000 | 1.0000 | 1.0000 |
| Fruit360 | Color | 0.5613 | 0.7305 | 0.5746 | 0.7673 | **0.6654** | **0.8821** |
| | Species | 0.4370 | 0.7552 | 0.5364 | 0.7631 | **0.6123** | **0.8504** |
| Card | Order | 0.3518 | 0.8458 | 0.3518 | 0.8458 | **0.3921** | **0.8842** |
| | Suits | 0.2711 | 0.6123 | 0.2711 | 0.6123 | **0.3104** | **0.7941** |
| CMUface | Emotion | 0.1576 | 0.6532 | 0.1590 | 0.6619 | **0.2053** | **0.8527** |
| | Glass | 0.2905 | 0.6869 | 0.4686 | 0.7505 | **0.4870** | **0.8324** |
| | Identity | 0.1998 | 0.6388 | 0.2677 | 0.7545 | **0.7441** | **0.9834** |
| | Pose | 0.4088 | 0.6473 | 0.4691 | 0.6409 | **0.5923** | **0.8736** |
| Stanford Cars | Color | 0.6539 | 0.8237 | 0.6830 | 0.8642 | **0.7533** | **0.9387** |
| | Type | 0.6207 | 0.7931 | 0.6429 | 0.8456 | **0.6616** | **0.8792** |
| Flowers | Color | 0.5653 | 0.7629 | 0.5828 | 0.7836 | **0.6940** | **0.8843** |
| | Species | 0.5620 | 0.7553 | 0.6019 | 0.7996 | **0.6724** | **0.8719** |
| CIFAR-10 | Type | 0.4935 | 0.6741 | 0.5087 | 0.7102 | **0.5271** | **0.7394** |
| | Environment | 0.4302 | 0.6507 | 0.4643 | 0.6801 | **0.4828** | **0.7096** |

multiple salient embedding matrices and multiple clusterings therein; **AugDMC** Yao et al. [2023] leverages data augmentations to automatically extract features related to different aspects of the data using a self-supervised prototype-based representation learning method; **DDMC** Yao and Hu [2024] combines disentangled representation learning with a variational Expectation-Maximization (EM) framework; **Multi-MaP** Yao et al. [2024] relies on a contrastive user-defined concept to learn a proxy better tailored to a user's interest. It is worth noting that, in our experiments, we apply both traditional and deep learning baselines. Traditional methods rely on hand-crafted features, while deep learning methods directly utilize the original images as input.

**Hyperparameter** For each user's preference, we train the model for $1000$ epochs using Adam optimizer with a momentum of $0.9$. We tune all the hyper-parameters based on the loss score of Multi-Sub, where the learning rate is selected from $\{1e-1, 5e-2, 1e-2, 5e-3, 1e-3, 5e-4\}$, weight decay is chosen from $\{5e-4, 1e-4, 5e-5, 1e-5, 0\}$ for all the experiments. Most methods obtain each clustering by applying k-means Lloyd [1982] to the newly learned representations, while ours is end-to-end. The experiments are performed on four NVIDIA GeForce RTX 2080 Ti GPUs.

**Evaluation metrics** Considering the randomness of k-means for those applicable baselines, we run k-means 10 times and report the average clustering performance using two metrics, namely, Normalized Mutual Information (NMI) White et al. [2004] and Rand index (RI) Rand [1971]. These metrics range from $0$ to $1$ with higher value indicating better performance compared to the groundtruth.

## 4.1 Performance Comparison

Table 2 reports the clustering results. During the clustering stage, after we obtain the proxy word embedding of each image for a desired concept, we can concatenate the image embedding and the token embedding of proxy word. The results show that Multi-Sub consistently outperforms the baselines, demonstrating the superiority of the proposed method. This also indicates a strong generalization ability of the pre-trained model by CLIP, which can capture the features of data from different perspectives.

Our methodology uses the CLIP encoder and GPT-4 to derive clustering results, prompting an evaluation of their performance in a zero-shot manner. We introduce two zero-shot variants of CLIP: CLIP$_{GPT}$ and CLIP$_{label}$. CLIP$_{GPT}$ uses GPT-4 to generate candidate labels and performs zero-shot classification, while CLIP$_{label}$ uses ground truth labels directly, providing an optimal setting. As shown in Table 3, CLIP$_{label}$ generally outperforms CLIP$_{GPT}$ due to its use of accurate labels, while CLIP$_{GPT}$ introduces noise. Both variants perform equally on the Card dataset as GPT-4's labels match the groundtruth. Multi-Sub surpasses CLIP$_{GPT}$ and even outperforms CLIP$_{label}$ in all cases, demonstrating its ability to capture user-interest-based data aspects and confirming its efficacy. This superiority can be attributed to Multi-Sub's proxy word learning mechanism, which automatically adjusts textual

Table 4: Comparison of different text encoders. The significantly best results with 95% confidence are in bold.

| Dataset | Clustering | CLIP | | ALIGN | | BLIP | |
|---|---|---|---|---|---|---|---|
| | | NMI↑ | RI↑ | NMI↑ | RI↑ | NMI↑ | RI↑ |
| Fruit360 | Color | 0.6654 | 0.8821 | **0.7031** | **0.8925** | 0.6522 | 0.8814 |
| | Species | 0.6123 | 0.8504 | **0.6426** | **0.8565** | 0.6254 | 0.8536 |
| Card | Order | 0.3921 | 0.8842 | **0.4316** | **0.9023** | 0.3845 | 0.8359 |
| | Suits | 0.3104 | 0.7941 | **0.3226** | **0.8006** | 0.3151 | 0.7956 |
| CMUface | Emotion | 0.2053 | 0.8527 | **0.2148** | **0.8553** | 0.2081 | 0.8535 |
| | Glass | 0.4870 | 0.8324 | **0.4951** | **0.8351** | 0.4951 | **0.8353** |
| | Identity | 0.7441 | **0.9834** | **0.7514** | 0.9828 | 0.6853 | 0.8321 |
| | Pose | 0.5923 | 0.8736 | **0.6137** | **0.8942** | 0.5732 | 0.8427 |
| Standford Cars | Color | 0.7533 | **0.9387** | **0.7624** | 0.8942 | 0.5732 | 0.8427 |
| | Type | 0.6616 | 0.8792 | **0.6712** | **0.8865** | 0.6581 | 0.8731 |
| Flowers | Color | **0.694** | **0.8843** | 0.6925 | 0.8812 | 0.6843 | 0.8789 |
| | Species | **0.6724** | **0.8719** | 0.6693 | 0.8691 | 0.6627 | 0.8654 |
| CIFAR-10 | Type | 0.5271 | 0.7394 | **0.5342** | **0.7456** | 0.5221 | 0.7381 |
| | Environment | **0.4828** | **0.7096** | 0.4793 | 0.7064 | 0.4752 | 0.7038 |

Table 5: Ablation study of Multi-Sub. The results that achieved the highest and second highest performance for each clustering are indicated by boldface and underlined numerals, respectively.

| Dataset | Clustering | Subspace | clustering with $h(\phi(w_i))$ | | | | | | clustering with $\mathbf{t}_i^*$ | | | | clustering with $\phi(w_i)$ | | | |
|---|---|---|---|---|---|---|---|---|---|---|---|---|---|---|---|---|
| | | | image | | text | | concatenate | | text | | concatenate | | text | | concatenate | |
| | | | NMI↑ | RI↑ | NMI↑ | RI↑ | NMI↑ | RI↑ | NMI↑ | RI↑ | NMI↑ | RI↑ | NMI↑ | RI↑ | NMI↑ | RI↑ |
| CIFAR-10 | Type | $h(\phi(w_i))$ | 0.3649 | 0.6546 | 0.4789 | 0.6607 | 0.5208 | 0.7281 | 0.4586 | 0.6331 | 0.4987 | 0.6933 | 0.4438 | 0.6282 | 0.4996 | 0.7069 |
| | | $\mathbf{t}_i^*$ | 0.3581 | 0.6378 | 0.4634 | 0.6439 | 0.5114 | 0.7189 | 0.4704 | 0.6586 | 0.5136 | 0.7196 | 0.4672 | 0.6524 | 0.5013 | 0.7136 |
| | | $\phi(w_i)$ | 0.3715 | 0.6589 | 0.4737 | 0.6563 | 0.5185 | 0.7211 | 0.4601 | 0.6420 | 0.5033 | 0.6989 | 0.4821 | 0.6638 | **0.5271** | **0.7394** |
| | Envrionment | $h(\phi(w_i))$ | 0.4271 | 0.6764 | 0.4533 | 0.6813 | 0.4737 | 0.6905 | 0.4249 | 0.6537 | 0.4149 | 0.6662 | 0.4336 | 0.6691 | 0.4569 | 0.6836 |
| | | $\mathbf{t}_i^*$ | 0.4216 | 0.6677 | 0.4229 | 0.6533 | 0.4496 | 0.6630 | 0.4336 | 0.6689 | 0.4563 | 0.6781 | 0.4264 | 0.6596 | 0.4514 | 0.6695 |
| | | $\phi(w_i)$ | 0.4320 | 0.6837 | 0.4507 | 0.6762 | 0.4686 | 0.6834 | 0.4218 | 0.6541 | 0.4432 | 0.6631 | 0.4586 | 0.6876 | **0.4828** | **0.7096** |

embeddings based on user-defined interests, creating more accurate proxy word embeddings. This approach reduces noise compared to CLIP$_{GPT}$, which suffers from label mismatches. Additionally, Multi-Sub's iterative learning process refines these embeddings, optimizing alignment between text and image representations.

## 4.2 Ablation study

**Different ways of constructing subspace** The subspace of the proposed method can be expanded by different embeddings, i.e., the token embedding of the proxy word $\phi(w_i)$, the text embedding of the proxy word $h(\phi(w_i))$, and the text embedding of the prompt $\mathbf{t}_i^* = h(\phi(t_i^*)\|\phi(w_i))$. These three kinds of embeddings can also be used to evaluate the clustering results in each case. In addition, we can use different combinations of learned embeddings (e.g., different concatenations of text and image embedding) as the final embedding for clustering. The results are shown in Table 5. It can be seen that using word token embedding usually achieves better results. This is expected since the word proxy directly reflects the image's category under the desired concept. The token word embedding subspace is also aligning well with CLIP's training method. In contrast, prompt embedding performs the worst as it introduces noise from user interest, dataset, and reference words, which are unnecessary for clustering. Additionally, most methods perform better when the same approach is used for constructing subspace and evaluating clustering results. Combining text and image embeddings generally enhances performance, capturing user interests from both aspects effectively.

**Effect of text encoder** Table 4 compares the performance of three text encoders—CLIP, ALIGN, and BLIP—across various datasets. The results indicate that ALIGN generally outperforms CLIP and BLIP in most tasks. This suggests that ALIGN's text encoder effectively captures and aligns textual and visual representations, enhancing clustering performance. ALIGN tends to excel in tasks that require distinguishing subtle visual differences influenced by textual descriptions, such as emotions and accessories in the CMUface dataset, and colors in the Fruit360 dataset. CLIP shows a strong tendency in identity-related tasks and complex object categorization, as evidenced by its performance in the CMUface identity task and Standford Cars type clustering. BLIP, while competitive, seems to perform better in categorical distinctions rather than abstract attributes, performing relatively well in species-related tasks across various datasets. These findings underscore the importance of effective text embeddings in multi-modal clustering frameworks.

We conducted an additional analysis using the Maximum Mean Discrepancy (MMD) metric to quantify the differences in the feature spaces generated by different text encoders (i.e., CLIP, ALIGN, and BLIP) in Table 6. The MMD results indicate that although our text prompts are simple, the

feature spaces generated by different text encoders exhibit significant distributional differences. The effectiveness of a text encoder can vary depending on the specific clustering task. For example, ALIGN tends to excel in tasks with more abstract attributes, such as colors and emotions, while CLIP shows strong performance in identity-related tasks. This variability underscores the importance of selecting an appropriate text encoder based on the specific application requirements. The difference between text encoders may come from the different corresponding pre-training tasks and this will be an interesting future direction.

Table 6: MMD between different text encoders across datasets.

| Dataset | Clustering | CLIP vs. ALIGN | CLIP vs. BLIP | ALIGN vs. BLIP |
|---|---|---|---|---|
| Fruit360 | Color | 0.234 | 0.198 | 0.211 |
| | Species | 0.189 | 0.172 | 0.183 |
| Card | Order | 0.215 | 0.202 | 0.219 |
| | Suits | 0.198 | 0.184 | 0.192 |
| CMUface | Emotion | 0.276 | 0.245 | 0.263 |
| | Glass | 0.231 | 0.217 | 0.225 |
| | Identity | 0.263 | 0.249 | 0.258 |
| | Pose | 0.245 | 0.228 | 0.239 |
| Stanford Cars | Color | 0.238 | 0.223 | 0.231 |
| | Type | 0.212 | 0.198 | 0.205 |
| Flowers | Color | 0.257 | 0.244 | 0.252 |
| | Species | 0.248 | 0.231 | 0.242 |
| CIFAR-10 | Type | 0.193 | 0.178 | 0.186 |
| | Environment | 0.178 | 0.162 | 0.174 |

**Visualization**  To further demonstrate the effectiveness of Multi-Sub, we visualize the representations from $CLIP_{label}$, $CLIP_{GPT}$, and Multi-Sub for color and species clustering tasks (Figure 3). In species clustering, $CLIP_{label}$ shows clear boundaries using ground truth labels, while $CLIP_{GPT}$ introduces noise from reference words. Multi-Sub outperforms both by effectively capturing image features and user interests with proxy word embeddings. In color clustering, both $CLIP_{label}$ and $CLIP_{GPT}$ focus on species features, resulting in less distinct clusters. Multi-Sub excels by clearly distinguishing colors, leveraging user-specific interests for improved alignment.  Overall, Multi-Sub consistently aligns embeddings with user interests, surpassing $CLIP_{label}$ and $CLIP_{GPT}$, demonstrating its robust multi-modal subspace proxy learning.

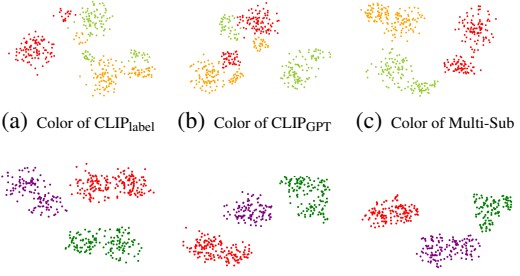

(a) Color of $CLIP_{label}$  (b) Color of $CLIP_{GPT}$  (c) Color of Multi-Sub

(d) Species of $CLIP_{label}$ (e) Species of $CLIP_{GPT}$ (f) Species of Multi-Sub

Figure 3: Visualization of feature embeddings and related labels on Fruit dataset. For the visualization of color, red, green, and yellow points indicate the color of red, green, and yellow, respectively. For the visualization of species, red, yellow, and purple points indicate the species of apple, banana, and grapes, respectively.

## 5   Conclusion and Limitations

In conclusion, our study mitigates an important challenge in multiple clustering: effectively identifying desired clustering results based on user interests or application purposes. We introduce Multi-Sub, a novel approach that integrates user-defined preferences into a customized multi-modal subspace proxy learning framework. By leveraging the synergy between CLIP and GPT-4, Multi-Sub automatically aligns textual prompts expressing user interests with corresponding visual representations. First, we observe reference words for user's interests from large language models. Given the absence of concrete class names in clustering tasks, our method uses these reference words to learn both text and vision embeddings tailored to user preferences. Extensive experiments across various visual multiple clustering tasks demonstrate that Multi-Sub consistently outperforms state-of-the-art techniques.

However, our approach has certain limitations. The reliance on large language models like GPT-4 can introduce biases inherent in these models, potentially affecting the clustering outcomes. Additionally, the field of multiple clustering lacks large, diverse datasets, which limits comprehensive evaluation. Although we have annotated CIFAR-10, more extensive datasets are needed.

# 6 Acknowledgment

Yao's research was funded in part by J.P. Morgan Chase & Co and Advata Gift funding. Any views or opinions expressed herein are solely those of the authors listed, and may differ from the views and opinions expressed by J.P. Morgan Chase & Co. or its affiliates. This material is not a product of the Research Department of J.P. Morgan Securities LLC. This material should not be construed as an individual recommendation for any particular client and is not intended as a recommendation of particular securities, financial instruments or strategies for a particular client. This material does not constitute a solicitation or offer in any jurisdiction.

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

# A  Appendix

## A.1   Further Analysis

**Parameters Analysis.**   To show the sensitivity of the balancing factor $\lambda$ that is the only hyper-parameter in our proposed method, the experiments were conducted on CIFAR-10. We varied the value of $\lambda$ from 0.1 to 1.0 in increments of 0.1 to observe its effect on the model's performance. As shown in Fig. 4, we can observe that different values of $\lambda$ can work with our method and the optimal performance for "Type" & "Environment" clustering is achieved when $\lambda$ is set to 0.5. When $\lambda$ is too low, the model focuses too much on maximizing the distances between different clusters, which can lead to less cohesive clusters. Conversely, when $\lambda$ is too high, the model emphasizes compactness within clusters at the expense of inter-cluster separation, leading to less distinct clusters. Therefore, we set $\lambda$ to be 0.5 for all datasets, which confirms the robustness of our method.

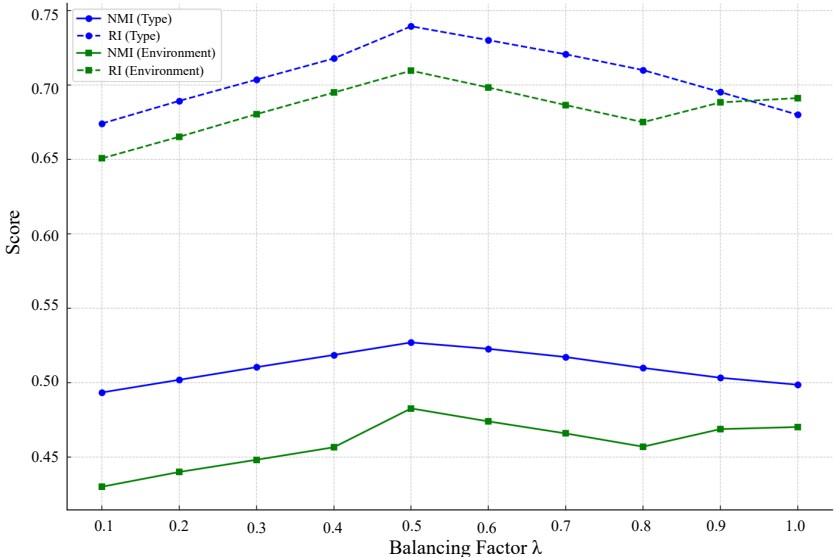

Figure 4: Sensitivity analysis of balancing factor $\lambda$ on CIFAR-10 dataset.

**Model Adaptability.**   To evaluate how Multi-Sub adapts to new user demands not originally provided in the ground-truth of the dataset, we conducted an additional experiment using the Fruit dataset. Specifically, we introduced a new demand based on the shape of the fruits, with the prompt set as "fruit with the shape of *". We categorized the fruits into two shapes, that is, round and elongated. Although this specific demand may not be common in practical applications, it serves as an exploratory experiment to test the adaptability of our method.

The results in Table 7 demonstrate that Multi-Sub successfully adapted to the new user demand of shape. The model learned to align the textual descriptions of shapes with the visual features, resulting in a clustering under the new subspace of shape.

Table 7: Clustering performance based on shape demand on the Fruit dataset.

| Method | NMI | RI |
|---|---|---|
| MSC | 0.553 | 0.762 |
| MCV | 0.586 | 0.787 |
| ENRC | 0.603 | 0.825 |
| iMClusts | 0.629 | 0.821 |
| AugDMC | 0.643 | 0.844 |
| Multi-MaP | 0.717 | 0.875 |
| Multi-Sub (ours) | **0.752** | **0.891** |

