# OpenReview forum: "Customized Multiple Clustering via Multi-Modal Subspace Proxy Learning"
_NeurIPS.cc/2024/Conference — NeurIPS 2024 poster_

### Official Review · Reviewer_r2bm · 2024-07-01

**Soundness:** 3
**Presentation:** 3
**Contribution:** 3
**Rating:** 5
**Confidence:** 4

**Summary:**

This paper incorporates a multi-model subspace proxy learning (Multi-Sub) to design a novel end-to-end multiple clustering approach and utilizes the synergistic capabilities of CLIP and GPT-4 to align textual prompts expressing user preferences with corresponding visual representations. The main contributions of Multi-Sub can be summarized as follows:
1. Capturing user’s clustering interest: Existing works struggle to adapt to diverse user-specific needs in data grouping. To overcome this limitation, Multi-Sub explicitly captures a user’s clustering interest by learning the desired clustering proxy under a user’s interest and aligning textual interest with corresponding visual features.
2. Simultaneous optimized framework: The previous methods separated the representation learning and clustering stages. Different from them, Multi-Sub obtains both the desired representations and clustering simultaneously, which significantly improve the clustering performance and efficiency.
3. Extensive experimental validation: Extensive experiments on all public multiple clustering tasks demonstrate that Multi-Sub outperform other methods. Moreover, a series of ablation studies further verify the effectiveness of Multi-Sub.

**Strengths:**

1. In real world, data may have multiple aspects that they can be grouped into different clusters. However, existing methods solely consider a single partition. So, it is meaningful to propose an effective algorithm to overcome this problem.
2. The authors leveraged large language models (LLMs), including GPT-4 and CLIP, to align image and textual representations in the same subspace. Then, multi-modal subspace proxy learning is introduced to allow for the customized representation of data in terms specific to the user’s interests.
3. Experimental results on public datasets show that the Multi-Sub method has a significant improvement, indicating the effectiveness of the propose method.

**Weaknesses:**

1. To change the two-stage learning approach of previous works, Multi-Sub aims to learn representation and clustering simultaneously. However, Multi-Sub employs a two-phase iterative approach to align and cluster images in training process, including (1) Phase I: Learning and Alignment; (2) Phase II: Clustering and Optimization. I wonder if this is another form of two-stage task.
2. The description of Clustering Loss is not very clear in Section 3.4, how to determine that samples belong to the same class? By pseudo-label? Where did the pseudo-label come from?
3. In this paper, the authors introduced large language models (LLMs) to learn representations and bridge the gap of textual and image features. But does the direct use of a pre-trained large language model introduce a priori information about the category, which can lead to unsupervised scenarios being corrupted?

**Questions:**

Please refer to the Weaknesses.

**Limitations:**

The authors have discussed social impacts and limitations.

---

> ### Author Rebuttal · Authors · 2024-08-07
>
> ### **W1: I wonder if this is another form of two-stage task.**
>
> Thanks for your invaluable feedback. We will make it clear in the revision as follows. A two-stage process used by previous methods separates the representation learning and clustering entirely, where the representation learning is fully completed at first. This separation, however, can lead to sub-optimal clustering results, as the learned representations may not be fully aligned with the clustering objective. Instead, we obtain both the proxy word and the clustering alternatively. Concretely, we first learn the proxy word in a user-preferred subspace (i.e., **Eqn. (5)** in the submission). Then, we fix the proxy word and open the image encoder to obtain better image representations considering the clustering objective (i.e., the clustering loss at **line 240** in the submission). These two steps are repeated until convergence. Therefore, these two components can improve each other iteratively, which shows better performance than the separated two-stage strategy in previous methods.
>
> ---
>
> ### **W2: The description of Clustering Loss is not very clear in Section 3.4, how to determine that samples belong to the same class? By pseudo-label? Where did the pseudo-label come from?**
>
> Thanks for pointing this out. During the clustering, the proxy word is fixed, which can be used to obtain the pseudo-labels to determine the inter-cluster and intra-cluster relationships. Concretely, an offline clustering algorithm (e.g., k-means) can be applied to the currently fixed proxy words to group the samples as the pseudo-labels. We will clarify it in the subsequent versions.
>
> ---
>
> ### **W3: Does the direct use of a pre-trained large language model introduce a priori information about the category, which can lead to unsupervised scenarios being corrupted?**
>
> Thanks for your insightful comments. It should be noted that although the category information released by the LLM can help determine a better subspace for the proxy learning, it is hard to cover all ground-truth labels, especially for user-specific domains as in this work. The provided categories by the LLM can be too narrowed, too expansive, or without any overlapping. Therefore, the corresponding categories existing in the data are unknown or uncertain. Moreover, the exact label for each instance is still unknown as in most unsupervised scenarios. We will make it clear in the revision.

---

> > ### Comment · Reviewer_r2bm · 2024-08-13
> >
> > Thanks for the response. Some of my concerns have been addressed. After reading the responses and the comments from other reviewers, I would maintain the original score.

---

> > > ### Author Response · Authors · 2024-08-13
> > >
> > > Many thanks again for your insightful comments for improving our work. If applicable, your further suggestion on any remaining issues would be greatly appreciated.

---

### Official Review · Reviewer_vBFT · 2024-07-11

**Soundness:** 4
**Presentation:** 4
**Contribution:** 4
**Rating:** 7
**Confidence:** 5

**Summary:**

This paper presents an innovative approach for addressing the limitations of existing multiple clustering methods. By leveraging the synergistic capabilities of CLIP and GPT-4, Multi-Sub aligns textual prompts with visual representations to cater to diverse user-specific clustering needs. This method introduces a novel multi-modal subspace proxy learning framework, which automatically generates proxy words from large language models to represent data in terms specific to user interests. The experimental results demonstrate that Multi-Sub consistently outperforms existing baselines across various datasets. Overall, I believe this paper makes a substantial contribution to the field of deep clustering and holds significant practical application value.

**Strengths:**

The paper offers several notable strengths that contribute to its overall impact and significance in the field of multiple clustering:
1.	The integration of CLIP and GPT-4 for multi-modal subspace proxy learning is novel and effectively addresses the limitations of traditional multiple clustering methods.
2.	Multi-Sub excels in capturing and responding to diverse user interests, providing tailored clustering results without requiring extensive manual interpretation. Moreover, the performance gains come at a low cost and seem relatively easy to achieve.
3.	The writing is clear and easy to follow. The figures are well-drawn, allowing for a quick understanding of the research motivation and methodological design.
4.	Extensive experiments on a wide range of publicly available datasets demonstrate the robustness and generalizability of the proposed method.

**Weaknesses:**

Despite its strengths, there are some areas where the paper could be improved to enhance its clarity and applicability:
1. Although the paper mentions the hyperparameters used, a more detailed analysis and discussion on the sensitivity of the method to these parameters would be beneficial.
2. Given the method's iterative nature and the use of large models, there is a risk of overfitting, especially on smaller datasets. I am curious whether regularization techniques were used to address this issue?
3. Table 3 compares the impact of different text encoders on performance. Clearly, there are significant performance differences when using different encoders, and the authors have indeed analyzed this issue. However, I believe the reasons behind this phenomenon could be explored in depth. Intuitively, given that the input text is quite simple, the overall performance should not be particularly sensitive to the choice of text encoder.

**Questions:**

1. Although the paper mentions the hyperparameters used, a more detailed analysis and discussion on the sensitivity of the method to these parameters would be beneficial.
2. Given the method's iterative nature and the use of large models, there is a risk of overfitting, especially on smaller datasets. I am curious whether regularization techniques were used to address this issue?
3. Table 3 compares the impact of different text encoders on performance. Clearly, there are significant performance differences when using different encoders, and the authors have indeed analyzed this issue. However, I believe the reasons behind this phenomenon could be explored in depth. Intuitively, given that the input text is quite simple, the overall performance should not be particularly sensitive to the choice of text encoder.

**Limitations:**

The limitations are discussed in the paper by the authors. There is no potential negative societal impact

---

> ### Author Rebuttal · Authors · 2024-08-07
>
> ### **W1:  A more detailed analysis and discussion on the sensitivity of the method to these parameters.**
>
> We greatly appreciate your suggestion. To show the sensitivity of the balancing factor $\lambda$ that is the only hyper-parameter in our proposal, the experiments were conducted on CIFAR-10. We varied the value of $\lambda$ from 0.1 to 1.0 in increments of 0.1 to observe its effect on the model's performance. As shown in **Figure 1** of the one-page PDF, we can observe that different values of $\lambda$ can work with our method and the optimal performance for "Type" \& "Environment" clustering is achieved when $\lambda$ is set to 0.5. When $\lambda$ is too low, the model focuses too much on maximizing the distances between different clusters, which can lead to less cohesive clusters. Conversely, when $\lambda$ is too high, the model emphasizes compactness within clusters at the expense of inter-cluster separation, leading to less distinct clusters. Therefore, we set $\lambda$ to be 0.5 for all datasets, which confirms the robustness of our method.
>
> ---
>
> ### **W2: Whether regularization techniques were used to address overfitting?**
>
> We would like to clarify that the primary objective of our method is clustering rather than supervised learning where overfitting is a more prevalent issue. Specifically, clustering focuses on finding inherent patterns and structures in the data rather than fitting to specific target labels. Therefore, the regularization in clustering helps constrain the data subspace to observe data groups rather than alleviate the overfitting of a model. In this work, we have subspace under a user's preference in the proposed method for regularization. We will make this clear in the revision.
>
> ---
>
> ### **W3: There are significant performance differences when using different encoders, this phenomenon could be explored in depth.**
>
> Thanks for your insightful comments. We conducted an additional analysis using the Maximum Mean Discrepancy (MMD) metric to quantify the differences in the feature spaces generated by different text encoders (i.e., CLIP, ALIGN, and BLIP).
>
> | **Dataset**       | **Clustering** | **CLIP vs. ALIGN** | **CLIP vs. BLIP** | **ALIGN vs. BLIP** |
> | ----------------- | :------------: | :----------------: | :---------------: | :----------------: |
> | **Fruit360**      | Color          | 0.234              | 0.198             | 0.211              |
> |                   | Species        | 0.189              | 0.172             | 0.183              |
> | **Card**          | Order          | 0.215              | 0.202             | 0.219              |
> |                   | Suits          | 0.198              | 0.184             | 0.192              |
> | **CMUface**       | Emotion        | 0.276              | 0.245             | 0.263              |
> |                   | Glass          | 0.231              | 0.217             | 0.225              |
> |                   | Identity       | 0.263              | 0.249             | 0.258              |
> |                   | Pose           | 0.245              | 0.228             | 0.239              |
> | **Stanford Cars** | Color          | 0.238              | 0.223             | 0.231              |
> |                   | Type           | 0.212              | 0.198             | 0.205              |
> | **Flowers**       | Color          | 0.257              | 0.244             | 0.252              |
> |                   | Species        | 0.248              | 0.231             | 0.242              |
> | **CIFAR-10**      | Type           | 0.193              | 0.178             | 0.186              |
> |                   | Environment    | 0.178              | 0.162             | 0.174              |
>
> The MMD results indicate that although our text prompts are simple, the feature spaces generated by different text encoders exhibit significant distributional differences.
>
> The effectiveness of a text encoder can vary depending on the specific clustering task. For example, ALIGN tends to excel in tasks with more abstract attributes, such as colors and emotions, while CLIP shows strong performance in identity-related tasks. This variability underscores the importance of selecting an appropriate text encoder based on the specific application requirements. The difference between text encoders may come from the different corresponding pre-training tasks and this will be an interesting future direction.

---

> > ### Comment · Reviewer_vBFT · 2024-08-12
> >
> > I appreciate the authors' detailed responses to my feedback. After carefully reviewing the rebuttal, I am satisfied that they have fully addressed all of my concerns. The datasets used in this work is quite comprehensive, and the detailed description provided so far has greatly helped my understanding. Overall, the Multi-Sub shows strong potential for user-friendly clustering tasks, which I find highly valuable. The method's contribution is clear: it effectively aligns user needs with image-level features by using textual information as a bridge, all within an end-to-end training process that integrates both textual representations and clustering-oriented fusion.
> >
> > In light of the novelty of the proposed method and the improvements made, I'm raising my rating to 7.

---

> > > ### Author Response · Authors · 2024-08-12
> > >
> > > We are very grateful for your appreciation and endorsement. Your feedback holds significant value in helping us enhance our work. We will carefully incorporate them in our revision.

---

### Official Review · Reviewer_Evc4 · 2024-07-12

**Soundness:** 3
**Presentation:** 3
**Contribution:** 3
**Rating:** 5
**Confidence:** 4

**Summary:**

The paper is about Multiple Clustering, which is an interesting topic. The authors propose a novel end-to-end multiple clustering approach that incorporates a multi-modal subspace proxy learning framework. The paper is well written and well organized. However, there are several concerns in the current version of the paper that addressing them will increase the quality of this paper.

**Strengths:**

1 The authors' idea of using large models to aid clustering is novel.
2 The paper is clearly structured and easy to understand.
3 The paper has sufficient experiments to support its point of view.

**Weaknesses:**

1 The authors point out that different clustering results can be given for different customization needs of users. Then it will bring several associations (not necessarily accurate): a. What should be done if the user's demand is exactly opposite to the potential clustering distribution? b. The experiments do give different clustering results for different demand types, if the user proposes a new type of demand, can the model also adaptively adjust?
2 Figure 2 is well drawn but could be further improved, some icons and fonts need to be adjusted.
3 The authors point out that their model is capable of outputting clustering results directly, and then there should be a corresponding formula to represent this. In addition, it is hoped that the authors will discuss further why, if it is not a difficult task to output clustering results directly, few previous methods have done so.
4 Authors should add details about the dataset, such as data size, feature types, etc.

**Questions:**

Considering that the authors did not add an appendix, are there any other discussions or experiments?

**Limitations:**

Yes.

---

> ### Author Rebuttal · Authors · 2024-08-07
>
> ### **W1: Several associations.**
>
> Thanks for your insightful comments. Multi-Sub works by learning in the user-preferred subspace. Therefore, it is theoretically unlikely that the learned representations are completely opposite to the user's demand under such aligned subspace. We will make it clear in the revision.
>
> To evaluate how Multi-Sub adapts to new user demands not originally provided in the ground-truth of the dataset, we conducted an additional experiment using the Fruit dataset. Specifically, we introduced a new demand based on the shape of the fruits, with the prompt set as "fruit with the shape of *". We categorized the fruits into two shapes, that is, round and elongated. Although this specific demand may not be common in practical applications, it serves as an exploratory experiment to test the adaptability of our method.
>
> The following results demonstrate that Multi-Sub successfully adapted to the new user demand of shape. The model learned to align the textual descriptions of shapes with the visual features, resulting in a clustering under the new subspace of shape. Thanks for the suggestion and we will include this experiment in the revision.
>
> | **Method**        | **NMI** | **RI**  |
> | ----------------- | ------- | ------- |
> | MSC               | 0.553   | 0.762   |
> | MCV               | 0.586   | 0.787   |
> | ENRC              | 0.603   | 0.825   |
> | iMClusts          | 0.629   | 0.821   |
> | AugDMC            | 0.643   | 0.844   |
> | Multi-MaP         | 0.717   | 0.875   |
> | **Multi-Sub (ours)** | **0.752** | **0.891** |
>
> ---
>
> ### **W2: Figure 2.**
>
> Thanks for pointing this out and we will carefully revise Figure 2 in the revision.
>
> ---
>
> ### **W3: Clustering Loss.**
>
> Thanks for the suggestion and we will make the following clear in the revision. Previous methods often use a two-stage process that they separate representation learning and clustering to simplify the optimization process. This separation, however, can lead to sub-optimal clustering results, as the learned representations may not be fully aligned with the clustering objective. Instead, we obtain both the proxy word and the clustering alternatively and simultaneously. Concretely, we first learn the proxy word in a user-preferred subspace (i.e., Eqn. (5) in the submission). Then, we fix the proxy word and open the image encoder to obtain better image representations considering the clustering objective (i.e., the clustering loss at line 240 in the submission). These two steps are repeated alternatively until convergence.
>
>
> ---
>
> ### **W4: Datasets.**
>
> Thanks for the suggestion. We will make the following descriptions clear in the revision.
>
> 1. **Stanford Cars**: This dataset contains 1,200 images of cars annotated with labels for both color and type (e.g., sedan, SUV).
> 2. **Card**: This dataset includes 8,029 images of playing cards with labels for rank (Ace, King, Queen, etc.) and suits (clubs, diamonds, hearts, spades).
> 3. **CMUface**: This dataset consists of 640 facial images with annotations for pose, identity, glasses, and emotions.
> 4. **Flowers**: This dataset includes 1,600 flower images labeled by color and species (e.g., iris, aster).
> 5. **Fruit**: This dataset comprises 105 images of fruits with labels for species (apples, bananas, grapes) and color (green, red, yellow).
> 6. **Fruit360**: Similar to the Fruit dataset, Fruit360 contains 4,856 images of various fruits with detailed annotations for species and color.
> 7. **CIFAR-10**: This dataset is structured to have 60,000 images clustered based on type (transportation, animals) and environment (land, air, water).
>
> The data size, handcrafted features, and clusters for all datasets we used are also summarized in the following table. It is worth noting that, in our experiments, we apply both traditional and deep learning baselines. Traditional methods rely on hand-crafted features, while deep learning methods directly utilize the original images as input.
>
> | **Datasets**    | **# Samples** |        **# Hand-crafted features**                  | **# Clusters** |
> | :-------------: | :-----------: | :--------------------------------------------: | :------------: |
> | Stanford Cars   |     1,200      | wheelbase length; body shape; color histogram  |      4;3       |
> | Card            |     8,029      | symbol shapes; color distribution              |     13;4       |
> | CMUface         |      640       | HOG; edge maps                                 |    4;20;2;4    |
> | Fruit           |      105       | shape descriptors; color histogram             |      3;3       |
> | Fruit360        |     4,856      | shape descriptors; color histogram             |      4;4       |
> | Flowers         |     1,600      | petal shape; color histogram                   |      4;4       |
> | CIFAR-10        |    60,000      | edge detection; color histograms; shape descriptors |    2;3      |
>
> ---
>
> ### **Question.**
>
> We do have some additional experiments as in the **one-page PDF** that we will add in the revision. Those experiments include:
>
> - Clustering performance based on shape demand on the Fruit dataset, demonstrating the adaptability of Multi-Sub for new demands not existing in ground-truth clusterings provided by the data.
> - Sensitivity analysis of the balancing factor $\lambda$ on the CIFAR-10 dataset, indicating a best value of 0.5, which is expected since when $\lambda$ is too low, the model focuses too much on maximizing the distances between different clusters, which can lead to less cohesive clusters. Conversely, when $\lambda$ is too high, the model emphasizes compactness within clusters at the expense of inter-cluster separation, leading to less distinct clusters. Therefore, 0.5 is used for all datasets.
> - Quantifying feature space differences generated by different text encoders using the Maximum Mean Discrepancy (MMD) metric, showing that even with simple text prompts in our work, different pre-trained text encoders vary in their abilities.

---

### Official Review · Reviewer_HfG7 · 2024-07-12

**Soundness:** 2
**Presentation:** 3
**Contribution:** 3
**Rating:** 6
**Confidence:** 4

**Summary:**

This paper introduces an end-to-end multi-clustering method that integrates a multimodal subspace proxy learning framework. It combines text prompts expressing user preferences with corresponding visual representations to achieve clustering based on user interests.

**Strengths:**

1.The clustering task, driven by user interests, aligns better with user preferences and is more applicable to real-world scenarios.
2.The experimental results are promising, and the methodology is clear and logical.

**Weaknesses:**

1.The contributions of the paper are not very clear. At first glance, it appears to merely combine CLIP and GPT, lacking innovative architecture.
2.The baseline methods chosen for comparison are neither cited nor introduced.
3.Section 5 discusses only limitations, lacking a discussion on broader impacts.

**Questions:**

The evaluation metrics mentioned in the paper require comparing results with ground truth values. How were the multiple clustering ground truth values in the dataset obtained? How is the accuracy of these ground truth values ensured?

**Limitations:**

The authors only address the limitations of their work and do not discuss broader impacts.

---

> ### Author Rebuttal · Authors · 2024-08-07
>
> ### **W1: The contributions of the paper.**
>
> Thank you for the suggestion. We will carefully emphasize our contribution in the revision as follows:
>
> Given only a high-level user interest in an unsupervised scenario without any class labels or names, we cannot directly apply CLIP. Instead, we must learn the unknown proxy word in a continuous space, which is challenging. The subspace approach proposed in this work aids effective learning. More importantly, our proposal can simultaneously optimize the proxy word and the clustering result. We will also emphasize this in our framework figure in the revision.
>
> ---
>
> ### **W2: The baseline methods chosen for comparison.**
>
> Thank you for pointing this out. We will provide detailed descriptions in the revision as follows:
>
> 1. **MSC** [2] is a traditional multiple clustering method that uses hand-crafted features to automatically find different feature subspaces for different clusterings.
> 2. **MCV** [1] leverages multiple pre-trained feature extractors as different views of the same data.
> 3. **ENRC** [3] integrates auto-encoder and clustering objective to generate different clusterings.
> 4. **iMClusts** [4] is a deep multiple clustering method that leverages the expressive representational power of deep autoencoders and multi-head attention to generate multiple salient embedding matrices and multiple clusterings therein.
> 5. **AugDMC** [6] leverages data augmentations to automatically extract features related to different aspects of the data using a self-supervised prototype-based representation learning method.
> 6. **DDMC** [5] combines disentangled representation learning with a variational Expectation-Maximization (EM) framework.
> 7. **Multi-MaP** [7] relies on a contrastive user-defined concept to learn a proxy better tailored to a user's interest.
>
> ---
>
> ### **W3: Broader impacts.**
>
> Thank you for the suggestion. We will add the discussion that our proposal has potential in various applications like personalized marketing. For example, it can enhance advertisement effectiveness by tailoring clustering results to business preferences.
>
> ---
>
>
> ### **Question: How were the multiple clustering ground truth values in the dataset obtained?**
>
> All multiple clustering datasets used in our experiments are public and come with pre-defined ground-truth labels in different aspects for evaluation that have been widely used. For datasets like Stanford Cars and CIFAR-10, the clustering ground truth labels are derived from their existing class labels. Below are the details of the datasets:
>
> 1. **Stanford Cars**: This dataset contains 1,200 images of cars annotated with labels for both color and type (e.g., sedan, SUV).
>
> 2. **Card**: This dataset includes 8,029 images of playing cards with labels for rank (Ace, King, Queen, etc.) and suits (clubs, diamonds, hearts, spades).
>
> 3. **CMUface**: This dataset consists of 640 facial images with annotations for pose, identity, glasses, and emotions.
>
> 4. **Flowers**: This dataset includes 1,600 flower images labeled by color and species (e.g., iris, aster).
>
> 5. **Fruit**: This dataset comprises 105 images of fruits with labels for species (apples, bananas, grapes) and color (green, red, yellow).
>
> 6. **Fruit360**: Similar to the Fruit dataset, Fruit360 contains 4,856 images of various fruits with detailed annotations for species and color.
>
> 7. **CIFAR-10**: This dataset is structured to have 60,000 images clustered based on type (transportation, animals) and environment (land, air, water).
>
> The data size, handcrafted features, and clusters for all datasets we used are also summarized in the table. It is worth noting that, in our experiments, we apply both traditional and deep learning baselines. Traditional methods rely on hand-crafted features, while deep learning methods directly utilize the original images as input.
>
> | **Datasets**    | **# Samples** |        **# Hand-crafted features**                  | **# Clusters** |
> | :-------------: | :-----------: | :--------------------------------------------: | :------------: |
> | Stanford Cars   |     1,200      | wheelbase length; body shape; color histogram  |      4;3       |
> | Card            |     8,029      | symbol shapes; color distribution              |     13;4       |
> | CMUface         |      640       | HOG; edge maps                                 |    4;20;2;4    |
> | Fruit           |      105       | shape descriptors; color histogram             |      3;3       |
> | Fruit360        |     4,856      | shape descriptors; color histogram             |      4;4       |
> | Flowers         |     1,600      | petal shape; color histogram                   |      4;4       |
> | CIFAR-10        |    60,000      | edge detection; color histograms; shape descriptors |    2;3      |
>
> ---
>
> #### **References**
> [1]. J. Guérin and B. Boots. Improving image clustering with multiple pretrained cnn feature extractors.In British Machine Vision Conference 2018, BMVC 2018, 2018.
>
> [2]. J. Hu, Q. Qian, J. Pei, R. Jin, and S. Zhu. Finding multiple stable clusterings. Knowledge and Information Systems, 2017.
>
> [3]. L. Miklautz, D. Mautz, M. C. Altinigneli, C. Böhm, and C. Plant. Deep embedded non-redundant clustering. In Proceedings of the AAAI conference on artificial intelligence, 2020.
>
> [4]. L. Ren, G. Yu, J. Wang, L. Liu, C. Domeniconi, and X. Zhang. A diversified attention model for interpretable multiple clusterings. IEEE Transactions on Knowledge and Data Engineering, 2022.
>
> [5]. J. Yao and J. Hu. Dual-disentangled deep multiple clustering. In Proceedings of the 2024 SIAM International Conference on Data Mining (SDM), 2024.
>
> [6]. J. Yao, E. Liu, M. Rashid, and J. Hu. Augdmc: Data augmentation guided deep multiple clustering. In INNS DLIA@IJCNN, 2023.
>
> [7]. J. Yao, Q. Qian, and J. Hu. Multi-modal proxy learning towards personalized visual multiple clustering. In Proceedings of the IEEE/CVF Conference on Computer Vision and Pattern Recognition 2024.

---

> > ### Comment · Reviewer_HfG7 · 2024-08-13
> >
> > Thank you for he response. All my concerns have been addressed, and I recognize the contributions of this paper. Therefore, I will raise my score.

---

> > > ### Author Response · Authors · 2024-08-13
> > >
> > > We highly appreciate your acknowledgment and will carefully take your insightful feedback to improve our revision.

---

### Author Rebuttal · Authors · 2024-08-07

Dear Reviewers,

We sincerely thank your invaluable time and efforts invested in reviewing our submission. Your constructive and insightful feedback are greatly appreciated for improving our revision.

We have carefully responded to all the questions and concerns raised in the individual rebuttal sections. Additionally, we have a one-page PDF that includes all the related tables/figures. Please kindly check if necessary.

Thank you once again for your insightful comments and suggestions, which have greatly contributed to the improvement of our work.

Best regards,

Authors

---

### Decision · Program_Chairs · 2024-09-25

**Decision:**

Accept (poster)

**Comment:**

This paper received 4 reviews from experts in the field. The paper received the following reviews:  1 Accept, 1 Weak Accept and 2 Borderline Accepts. The paper has clearly positive reviews and the decision is to accept. Please incorporate reviewer comments in your final camera ready submission.